# Theoretical Study on the Open-Shell Electronic Structure and Electron Conductivity of [18]Annulene as a Molecular Parallel Circuit Model

**DOI:** 10.3390/nano14010098

**Published:** 2023-12-31

**Authors:** Naoka Amamizu, Mitsuhiro Nishida, Keisuke Sasaki, Ryohei Kishi, Yasutaka Kitagawa

**Affiliations:** 1Department of Materials Engineering Science, Graduate School of Engineering Science, Osaka University, Toyonaka, Osaka 560-8531, Japan; mitsuhiro.nishida@cheng.es.osaka-u.ac.jp (M.N.); keisuke.sasaki@cheng.es.osaka-u.ac.jp (K.S.); kishi.ryohei.es@osaka-u.ac.jp (R.K.); 2Center for Quantum Information and Quantum Biology (QIQB), International Advanced Research Institute (IARI), Osaka University, Toyonaka, Osaka 560-0043, Japan; 3Research Center for Solar Energy Chemistry (RCSEC), Graduate School of Engineering Science, Osaka University, Toyonaka, Osaka 560-8531, Japan; 4Innovative Catalysis Science Division, Institute for Open and Transdisciplinary Research Initiatives (ICS-OTRI), Osaka University, Suita, Osaka 565-0871, Japan; 5Spintronics Research Network Division, Institute for Open and Transdisciplinary Research Initiatives (SRN-OTRI), Osaka University, Toyonaka, Osaka 560-8531, Japan

**Keywords:** molecular electronics, molecular parallel circuit, open-shell systems, density functional theory, elastic scattering Green’s function theory

## Abstract

Herein, the electron conductivities of [18]annulene and its derivatives are theoretically examined as a molecular parallel circuit model consisting of two linear polyenes. Their electron conductivities are estimated by elastic scattering Green’s function (ESGF) theory and density functional theory (DFT) methods. The calculated conductivity of the [18]annulene does not follow the classical conductivity, i.e., Ohm’s law, suggesting the importance of a quantum interference effect in single molecules. By introducing electron-withdrawing groups into the annulene framework, on the other hand, a spin-polarized electronic structure appears, and the quantum interference effect is significantly suppressed. In addition, the total current is affected by the spin polarization because of the asymmetry in the coupling constant between the molecule and electrodes. From these results, it is suggested that the electron conductivity as well as the quantum interference effect of π-conjugated molecular systems can be designed using their open-shell nature, which is chemically controlled by the substituents.

## 1. Introduction

Microfabrication technology for integrated circuits has been developed according to Moore’s law proposed in 1965 [1,2]. The current gate length of transistors has reached less than 10 nm [3]. This miniaturization technology of transistors has increased integration density, which drastically improves the performance of electron devices. On the other hand, it has been suggested that the miniaturization of silicon (Si)-based devices is almost reaching its limit. In addition, energy loss due to leakage currents induced by the miniaturization has been a problem. Recent developments in the fabrication technology of power semiconductors using silicon carbide (SiC) and gallium nitride (GaN) have succeeded in realizing higher efficiency and lower power consumption than those of Si-based semiconductors [4,5,6]. As another approach to the problem, molecular electronics, i.e., the use of single molecules for electronic components, has attracted much attention in terms of new nanomaterials as a “bottom-up approach” [7,8,9,10,11]. In 1974, a molecular rectifier was first proposed by Aviram and Ratner as molecular electronics [12]. A couple of decades after their proposal, techniques for measurements of single-molecule electron conductivity were developed [13,14,15,16], and various functional single-molecule components were proposed with these developments: molecular wires connecting individual single-molecule components [17,18,19,20], transistors or switches controlling two states with different conductivity [21,22,23,24,25], and diodes which show rectifier properties by connecting donor and acceptor parts [26,27,28,29]. Furthermore, it has become possible to realize logic operations by using single-molecule components [30,31,32,33]. On the other hand, through the development of computational procedures to simulate the properties of single molecules, the electron conductivities of various molecules have also been estimated by theoretical calculations [34,35,36,37,38,39]. For example, our group developed a method to simulate the electron conductivity of open-shell molecules by density functional theory (DFT) and elastic scattering Green’s function (ESGF) methods [40]. By applying the method to some linearly aligned polynuclear metal complexes, called extended metal atom chains (EMACs), we found a difference in electron conductivity between spin states, suggesting their potential for single-molecule transistors controlled by an external magnetic field [41,42,43].

For the use of these molecules as devices, one is required to integrate single-molecule components and to make molecular circuits. Recently, for example, integrated single-molecule components based on self-assembled monolayers (SAMs) have been proposed, demonstrating a variety of functionalities [44,45]. In addition, various molecular devices, such as a biosensor that detects molecular interactions, have been reported [46]. Interestingly, it has also been proposed that the non-linear quantum conduction of single molecules can be applied to neuromorphic computers because of its similarity to electrical signals in human brains [47,48].

In recent years, some interesting results on molecular circuits have been reported. For example, Ohm’s law, which indicates that the current value of a parallel circuit is equal to the total current through each resistor, is expected to fail in molecular parallel circuits because of quantum interference effects [49]. The conductance, G, in nano-scale parallel circuits of two components is given as
(1)$$G=G_{1}+G_{2}+2\sqrt{G_{1}G_{2}}\;,$$
where $G_{1}$ and $G_{2}$ are the conductance of respective components and $2\sqrt{G_{1}G_{2}}$ is the quantum interference term [49]. Vazquez et al. indicated that the conductance of a model molecule consisting of two parallel benzenes is larger than twice that of a single benzene, both experimentally and theoretically [50]. Recently, some exceptions to Equation (1) have been reported [51,52]; therefore, Equation (1) is still under discussion in terms of aromaticity, frontier orbital theory, and orbital interaction [51]. Concerning open-shell systems, the relationship between the conductivity of a molecular circuit, quantum interference, and open-shell nature has not yet been discussed.

From those points of view, in this study, we aimed to calculate the electron conductivity of molecular circuit models and to compare this with the conductivity expected from the classical Ohm’s law. We focused on [18]annulene as the molecular parallel circuit model, which was assumed to be a parallel circuit that can be divided into two (top and bottom) linear polyenes, as illustrated in Figure 1. The electronic structures and the electron conductivities of the model structures were calculated by density functional theory (DFT) and elastic scattering Green’s function (ESGF) methods.

## 2. Theoretical Background and Computational Details

### 2.1. Electron Conductivity Calculation of Single-Molecule

In this study, the electron conductivity of the single molecule, which had discrete orbitals, was calculated using the elastic scattering Green’s function proposed by Mujica and Luo [53,54,55]. As described below, some model molecules are open-shell systems; therefore, we used the extended Mujica and Luo method for the open-shell systems proposed by Nakanishi et al. [40]. In the extended Mujica and Luo method, the Hamiltonian of the system is written as
(2)$$H=H_{M}+H_{L}+H_{R}+U,$$
(3)$$H_{M}=\sum_{\sigma }\sum_{\alpha }E_{\alpha \sigma }^{0}\left|\left.\alpha_{\sigma }\right\rangle \right.\left\langle \left.\alpha_{\sigma }\right|\right.\equiv \sum_{\sigma }\sum_{\alpha }E_{\alpha \sigma }^{0}C_{\alpha \sigma I}^{*}C_{\alpha \sigma J}\left|\left.I\right\rangle \right.\left\langle \left.J\right|\right.,$$
(4)$$H_{L}=\sum_{\sigma }\sum_{i}E_{i\sigma }^{0}\left|\left.i_{\sigma }\right\rangle \right.\left\langle \left.i_{\sigma }\right|\right.,$$
(5)$$H_{R}=\sum_{\sigma }\sum_{j}E_{j\sigma }^{0}\left|\left.j_{\sigma }\right\rangle \right.\left\langle \left.j_{\sigma }\right|\right.,$$
(6)$$U=\sum_{\sigma }\sum_{I}\left(\sum_{i}\gamma_{iI,\sigma }\left|\left.i_{\sigma }\right\rangle \right.\left\langle \left.I\right|\right.+\sum_{j}\gamma_{jI,\sigma }\left|\left.j_{\sigma }\right\rangle \right.\left\langle \left.I\right|\right.\right)+complex\;conjugate,$$
where $H_{M}$, $H_{L}$, and $H_{R}$ are the Hamiltonians of the molecule, the left-sided, and right-sided electrodes, respectively; U is the interaction potential between the molecule and the electrodes; and $\gamma_{iI,\sigma }$ is the interaction between the *I*-th site of the molecule and the *i*-th orbital of the electrodes with spin *σ* (=α or β). Additionally, the transition operator is defined as
(7)T=U+UGU,
where G is the Green’s function,
(8)$$G\left(z\right)={\left(z-H\right)}^{-1}.$$

Here, z is a complex variable. Assuming that the electrodes only interact directly with the end-sites, i.e., site 1 and *N* of the molecule (see Figure 2) [53,54,55], the transition matrix can be written as
(9)$$T_{ij,\sigma }\left(z\right)=\gamma_{i1,\sigma }G_{1N,\gamma }\left(z\right)\gamma_{Nj,\sigma },$$
(10)$$G_{1N,\gamma }(z)=\sum_{\eta }\left\langle 1|\frac{1}{z-H}|\varphi_{\sigma }^{\eta }\right\rangle \left\langle \varphi_{\sigma }^{\eta }|N\right\rangle \;\;=\sum_{\eta }\frac{\left\langle 1|\varphi_{\sigma }^{\eta }\right\rangle \left\langle \varphi_{\sigma }^{\eta }|N\right\rangle }{z-\varepsilon_{\sigma }^{\eta }}\;\;\approx \sum_{\eta }\frac{\left\langle 1|\phi_{\sigma }^{\eta }\right\rangle \left\langle \phi_{\sigma }^{\eta }|N\right\rangle }{z-{\tilde{\varepsilon }}_{\sigma }^{\eta }},$$
where $\left|\left.\varphi_{\sigma }^{\eta }\right\rangle \right.$ is the eigenstate of the total Hamiltonian H ($H\left|\left.\varphi_{\sigma }^{\eta }\right\rangle \right.=\varepsilon_{\sigma }^{\eta }\left|\left.\varphi_{\sigma }^{\eta }\right\rangle \right.$). Since the end-sites are defined as the anchor atoms connected to the electrodes, the eigenstate, η, that overlaps with the end-sites only contributes to matrix elements in the Green’s function. Here, $\left|\left.\phi_{\sigma }^{\eta }\right\rangle \right.$ is approximated by the orbitals obtained from the Kohn–Sham equation for the finite systems consisting of the molecule sandwiched between the electrodes ($H\left|\left.\phi_{\sigma }^{\eta }\right\rangle \right.={\tilde{\varepsilon }}_{\sigma }^{\eta }\left|\left.\phi_{\sigma }^{\eta }\right\rangle \right.$) [56].

The conduction current density, $i_{\sigma }^{LR}$, of the system under the applied voltage $V_{D}$ by the electrodes is given as
(11)$$i_{\sigma }^{LR}=\frac{1}{2}\sum_{\eta }\frac{em_{e}k_{B}T}{2\pi^{2}\hbar^{3}}\int_{eV_{D}}^{\infty }{\left|T_{\sigma }\left(E\right)\right|}_{\eta }^{2}f_{\sigma }\left(E\right)dE,$$
(12)$$f_{\sigma }\left(E\right)=\left\{\ln\left[1+exp\left(\frac{E_{F,\sigma }+eV_{D}-E}{k_{B}T}\right)\right]-\ln\left[1+exp\left(\frac{E_{F,\sigma }-E}{k_{B}T}\right)\right]\right\},$$
where the constants e, $m_{e}$, $k_{B}$, and ℏ are the elementary charge, the electron mass, the Boltzmann constant, and the reduced Plank constant, respectively. T is the temperature of the system and $f_{\sigma }\left(E\right)$ is the Fermi distribution. $E_{F,\sigma }$ is the Fermi energy, defined as the intermediate value between the orbital energies of the HOMO and the LUMO of the extended molecule consisting of the molecule and the electrodes. Assuming that the interactions between the different scattering channels are negligible because the spacing between the molecular orbitals is large enough, then the transition probability is written as
(13)$${\left|T_{\sigma }\left(E\right)\right|}_{\eta }^{2}=\gamma_{L1,\sigma }^{2}\gamma_{NR,\sigma }^{2}\sum_{\eta }\frac{{\left|\left\langle 1|\phi_{\sigma }^{\eta }\right\rangle \right|}^{2}{\left|\left\langle \phi_{\sigma }^{\eta }|N\right\rangle \right|}^{2}}{{\left({\tilde{\varepsilon }}_{\sigma }^{\eta }-E\right)}^{2}+\Gamma_{\eta ,\sigma }^{2}},$$
where $\Gamma_{\eta ,\sigma }^{2}$ is the spin-dependent escape rate determined by Fermi’s golden rule,
(14)$$\Gamma_{\eta ,\sigma }=\frac{\gamma_{L1,\sigma }\left\langle 1|\phi_{\sigma }^{\eta }\right\rangle +\gamma_{NR,\sigma }\left\langle \phi_{\sigma }^{\eta }|N\right\rangle }{2}.$$

Here, $\left\langle 1|\phi_{\sigma }^{\eta }\right\rangle$ and $\left\langle \phi_{\sigma }^{\eta }|N\right\rangle$ denote the site-orbital overlap matrix elements between the end-sites and the extended molecule. Their product ${\left|\left\langle 1|\phi_{\sigma }^{\eta }\right\rangle \right|}^{2}{\left|\left\langle \phi_{\sigma }^{\eta }|N\right\rangle \right|}^{2}$ is called Site-overlap, which represents the delocalization of the molecular orbitals of the extended molecule; γ is the coupling constant, as explained below. Luo et al. proposed that the occupied molecular orbitals of electrodes interact with the LUMO of the molecule based on the frontier orbital theory [37,56]. Thus, the γ between the left electrode and site 1 is written as
(15)$$\gamma_{L1,\sigma }\left(LUMO\right)=V_{L,\sigma }\left(LUMO\right)d_{1,\sigma }^{*}\left(LUMO\right),$$
(16)$$V_{L,\sigma }^{2}\left(LUMO\right)=\frac{\left(\Delta E_{\sigma ,HOMO–LUMO}-\Delta E_{\sigma ,LUMO}\right)\Delta E_{\sigma ,LUMO}}{2},$$
(17)$$d_{1,\sigma }^{2}\left(LUMO\right)=\frac{\sum_{i}c_{1,i,\sigma }^{2}}{\sum_{a,i}c_{a,i,\sigma }^{2}},$$
where $V_{L,\sigma }\left(LUMO\right)$ is the interaction between the LUMO of the molecule and the HOMO of the electrode at spin σ, and $\Delta E_{\sigma ,LUMO}$ is the energy difference between them. $\Delta E_{\sigma ,HOMO–LUMO}$ is the HOMO–LUMO gap of the extended molecule and $d_{1,\sigma }^{2}(LUMO)$ is the ratio of the electron density at end-site 1 to that of the entire molecule. In other words, $V_{L,\sigma }\left(LUMO\right)$ expresses the strength of the junction between the molecule and the electrodes, $d_{1,\sigma }^{*}\left(LUMO\right)$ represents the degree of transmission from the electrode to the molecule, and their product, $\gamma_{L1,\sigma }\left(LUMO\right)$, determines the electron conductivity derived from the electron transfer between the electrodes and the molecule. Equations (15)–(17) can also be applied to the interaction between the end-site *N* and the right electrode with subscripts *N* and *R*. 

In this way, the current can be calculated by considering transition probabilities based on tunneling through the discrete molecular orbitals and their energies without an external electric field. The total conduction is $I^{LR}=Ai^{LR}$, where A is the effective injection area where the electrons transfer from the electrodes. Here, it is approximated that $A\approx \pi r_{s}^{2}$, where $r_{s}$ is the radius of a sphere with the same volume as an electron, $r_{s}={\left[3/\left(4\pi n\right)\right]}^{1/3}$, and *n* is the density of the electron, $n={\left(2m_{e}E_{F,\sigma }\right)}^{3/2}/\left(3{\pi^{2}\hbar }^{3}\right)$ [37]. Notably, the relationship between the electron conductivities and the quantum interference of the derivatives is discussed through the following formula,
(18)$$I=I_{1}+I_{2}+2\sqrt{I_{1}I_{2}},$$
instead of Equation (1) under the same voltage in this paper.

### 2.2. Computational Details

In this study, [18]annulene derivatives were used as models of the molecular parallel circuit, which was divided into two linear polyenes (top: Polyene A; bottom: Polyene B), as illustrated in Figure 1. The [18]annulene derivatives and the two polyenes were connected to the gold electrodes via thioketone groups, as illustrated in Figure 2. The anchor sulfur atoms were assumed to be connected to the bridge-site of the Au(111) surface, and the electrodes were approximated as gold dimers [40]. Several [18]annulene derivatives that involve electron-donating/withdrawing groups (X) were also examined together with the non-substituted (NS) derivative. Since some computational models show open-shell nature, as explained below, density functional theory (DFT) calculations with the spin-unrestricted approximation that enabled the spin polarization to be performed involved the static electron correlation effect. In addition, BHandHLYP [57], which is reported to be effective for spin-polarized open-shell systems, was utilized as the functional for all calculations [58,59].

These [18]annulene derivatives were first geometrically optimized by the BHandHLYP/6-31+G*(S), 6-31G*(other atoms) [57,60,61,62] level of theory, and then they were divided into the two polyenes (i.e., Polyene A and Polyene B). Only the hydrogen atoms added at the boundaries of the polyene models were optimized after the partition. The Cartesian coordinates of these model molecules are summarized in Appendix A. For these models, the electronic structures were obtained by BHandHLYP/LANL2DZ [63] (Au), 6-31+G* (S) and 6-31G* (other atoms) levels of theory. For all models, the charge and the spin states were assumed to be neutral and singlet, respectively.

The electron conductivity was calculated through ESGF methods using the DFT results, assuming that the temperature was 300 K and the intramolecular electron conduction only occurred through the molecular orbitals from LUMO+9 to HOMO−9 [40]. All DFT calculations were performed in the gas phase using Gaussian09 Rev. D01 [64]; the electron conductivity was simulated using our self-developed program.

**Figure 2 nanomaterials-14-00098-f002:**
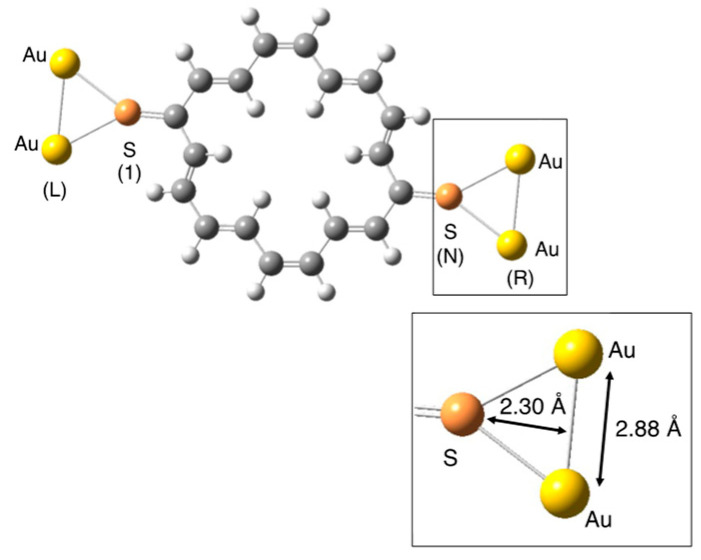
Calculated model systems. Electrodes are approximated by the gold dimers. The labels 1 and N represent the sulfur atoms at end-sites; L and R are the left and right electrodes, respectively (see the equations presented in Section 2.1). The distances between the middle point of the Au dimer and the sulfur atom were fixed at 2.30 Å, and the Au–Au distance was assumed to be 2.88 Å [37,65].

## 3. Results and Discussion

### 3.1. Conductivity of [18]Annulene: The Molecular Parallel Circuit Model

Figure 3 shows the current–voltage (*I*–*V*) characteristics of [18]annulene ($I_{\mathrm{annulene}}$). The current increased non-linearly with the increase in voltage, showing typical quantum electron conductivity characteristics. For the comparison, a total current of two independent polyenes (Polyene A and Polyene B) ($I_{\mathrm{Polyene}\;A}+I_{\mathrm{Polyene}\;B}=I_{A+B}$) and an estimated current for quantum systems, i.e., $I_{A+B}$ plus the quantum interference (QI) term ($I_{\mathrm{Polyene}\;A}+I_{\mathrm{Polyene}\;B}+2\sqrt{I_{\mathrm{Polyene}\;A}I_{\mathrm{Polyene}\;B}}=I_{A+B+Q}$) were also plotted. As shown in the figure, the directly calculated *I*–*V* characteristics of the [18]annulene were larger than the simple sum ($I_{A+B}$), but smaller than the value including the QI term ($I_{A+B+Q}$). To consider the difference in the conductivity between the [18]annulene and the polyenes ($I_{\mathrm{annulene}}$ and $I_{A+B}$, $I_{A+B+Q}$), several parameters, i.e., total spin angular momentum ($\langle {\hat{S}}^{2}\rangle$) and the coupling constants such as $\gamma_{L1},\;\gamma_{NR}$, are summarized in Table 1. In addition, Site-overlap values of each molecular orbital are also shown in Table 2. The Site-overlap values indicate that the most effective orbital to conductivity is HOMO−5 in annulene, whose electron distribution is illustrated in Figure 4a. 

On the other hand, one can find significant difference in $\langle {\hat{S}}^{2}\rangle$ values between [18]annulene and polyenes. The $\langle {\hat{S}}^{2}\rangle$ values represent whether the molecule is the closed-shell ($\langle {\hat{S}}^{2}\rangle =0$) or open-shell system ($\langle {\hat{S}}^{2}\rangle \neq 0$) [66]. From the calculated $\langle {\hat{S}}^{2}\rangle$ values in Table 1, the [18]annulene is found to be a closed-shell system, whereas the divided polyenes are weak open-shell systems. Moreover, according to Table 1 and Table 2, the coupling constants and Site-overlap values of α and β orbitals are the same in the [18]annulene, while they are different in the divided polyenes. In the closed-shell systems, the α and β orbital distributions are spatially similar to each other. On the other hand, in the open-shell systems, the α and β orbitals show spatially different distributions, indicating spin-polarized electronic states. In other words, α and β orbitals tend to localize at the left and the right sides of the polyenes, respectively. As a result, the parameters of the α and β orbitals are the same in the [18]annulene that has the closed-shell electronic structure, in contrast, the divided polyenes show different parameter values between the α and β orbitals due to their open-shell electronic structures. Therefore, the results indicate that the electron is not easily transmitted through the molecules in the polyenes because of the spin polarization. Indeed, the parameters related to the conductivity are different between the left and right sides ($\gamma_{L1,\sigma }$ and $\gamma_{NR,\sigma }$); thus, the conductivity of the polyenes is decreased in comparison with that of the annulene. On the other hand, $I_{A+B+Q}$ that fully involves the QI term becomes larger than $I_{\mathrm{annulene}}$, suggesting that the quantum interference must be examined partially in the systems with open-shell nature. 

### 3.2. Effects of Substituents

As explained in the previous section, the electron conductivity and the QI term are affected by the open-shell nature of the molecules. The result suggests a possibility that the electron conductivity of molecules can be controlled by changing the degree of spin polarization with the introduction of the substituent groups (X). From this point of view, the relationship between the open-shell nature and the conductivity is examined by introducing electron-donating (X = -OCH_3_, -OH) and -withdrawing (X = -CN, -NO_2_) substituents to the [18]annulene and the divided polyenes. Here, a pair of substituents were introduced symmetrically to carbon atoms with a larger distribution in HOMO−5, which dominantly contributes to the Site-overlap value in the non-substituted (NS) annulene, as shown in Figure 4a. Figure 4b shows the current values of the substituted annulenes and polyenes at 1.0 V. The introduction of the substituents significantly decreases the conductivity for both annulenes and polyenes, regardless of the electron-donating/withdrawing groups. However, the relationship in conductivity between the annulenes and the sum of two polyenes is different between the electron-donating and -withdrawing groups. In the case of the electron-donating groups, the current values of the annulenes are close to the total current values of the polyenes, including the QI term ($I_{\mathrm{annulene}}\approx I_{A+B+Q}$), whereas they become comparable to the simple sum of the current values of the two polyenes in the case of the withdrawing groups ($I_{\mathrm{annulene}}\approx I_{A+B}$). This suggests that the effect of the QI term depends on the π electronic states in the annulene ring. 

To further understand the effect, we examined the $\langle {\hat{S}}^{2}\rangle$ values. In the case of the NS model and its derivatives with electron-donating groups (-OCH_3_ and -OH), the calculated $\langle {\hat{S}}^{2}\rangle$ values are zero, although their polyenes show non-zero values, as summarized in Table 3. Therefore, the NS annulene and its derivatives with electron-donating groups are closed-shell systems, while the corresponding polyenes are open-shell systems. In contrast, in the case of the derivatives with electron-withdrawing groups (-CN and -NO_2_), both the derivatives and their polyenes show non-zero $\langle {\hat{S}}^{2}\rangle$ values, indicating the open-shell electronic structures. In order to confirm the effect of the open-shell nature in detail, the $\gamma_{L1},\;\gamma_{NR}$ and the Site-overlap totals of each derivative are summarized in Table 4. In addition, the parameters related to the transition probability, such as the Site-overlap of each molecular orbital, the HOMO–LUMO gap, coupling constants, etc., are summarized in Appendix A; the orbital energy is shown in Appendix A. The closed-shell models, i.e., the NS and its derivatives with electron-donating groups, have almost the same coupling constants and Site-overlap values between the left and right sides, and between the α and β orbitals. In contrast, the other models with open-shell electronic structures show different values between the α and β orbitals, and asymmetry between the left and right-sides. For example, $\gamma_{L1},\;\gamma_{NR}$ values are asymmetrical reflecting the spin polarization of the open-shell electronic states because the α and β molecular orbitals tend to localize on the opposite sides of the molecule. Since the current is proportional to $\gamma_{L1,\sigma }^{2}\gamma_{NR,\sigma }^{2}$ and the Site-overlap values, as shown in Equation (13), in conclusion, such asymmetry is considered to decrease the total current. 

By summarizing these results, we can suggest that: 

(i)In the case where annulene derivatives have closed-shell electronic structures, the QI term becomes significant ($I_{\mathrm{annulene}}\approx I_{A+B+Q}$). (ii)In the case where the annulene derivatives have open-shell electronic structures, the QI term is suppressed; therefore, the current of the annulenes is close to the sum of the current values of two polyenes ($I_{\mathrm{annulene}}\approx I_{A+B}$). This is similar to Ohmic behavior. In other words, the spin polarization, i.e., a separation between α and β orbitals, avoids the quantum interference in the molecular parallel circuit, and it can be realized by the introduction of an electron-withdrawing group. Moreover, in such spin-polarized systems, the total current also decreased due to the asymmetry in the coupling constants and Site-overlap values.

Finally, a reason why the introduction of donating groups as well as both the electron-withdrawing groups decreases the conductivity in comparison with the NS model is explained here. As mentioned above, the total current of the annulene derivatives with the electron-withdrawing groups decreases due to the spin polarization. In contrast, a difference in the molecular orbital distribution, especially HOMO−5, becomes important for models with electron-donating groups. The distribution in HOMO−5 of the NS model and substituted annulenes of the electron-donating groups are shown in Figure 5. The introduction of the electron-donating groups suppresses a degree of delocalization of HOMO−5 that contributes to the conductivity. This decrease is explained as follows: the π orbital energies of the annulene ring are shifted by the substitution effect, so that the energy difference between the π orbitals of the annulene ring and junction moieties (-S-Au_2_) becomes larger. As a result, the Site-overlap values become small due to a decrease in conjugation between the ring and junction moieties.

## 4. Conclusions

In this study, we examined the electron conductivity of [18]annulene and its derivatives with electron-donating/-withdrawing groups as models for molecular parallel circuits. In the case of the NS model and its derivatives with electron-donating groups, the electronic structures are closed-shell. Furthermore, their current values are larger than those of the sum of the two divided polyenes, indicating the importance of quantum interference effect in these systems. In the derivatives with the electron-withdrawing groups, however, the quantum interference effect is suppressed by the spin polarization of the open-shell nature. 

Thus, it is suggested that the quantum interference effect of the π-conjugated molecular parallel circuit can be designed with an open-shell nature, which is chemically controlled by the substituent groups. The open-shell systems with spin polarization due to the strong electron correlation have been a focus because of their applicability to functional materials [67,68]. However, a relationship between their open-shell nature and conductivity has not been clarified. Concerning the electron conductivity, our group has examined the relationship between electronic/spin structures and the electron conductivity of one-dimensional polynuclear nickel complexes through theoretical calculations [41,42,43]. Such extended metal atom chains (EMACs) are considered to be promising candidates for the molecular wire [69,70]; however, we found that they become candidates for single molecular switches controlled by an external magnetic field just by introducing the open-shell nature [41,42,43]. Thus, open-shell systems have great potential for the development of highly functional single molecular devices, which cannot be realized by closed-shell systems. Recently, it has been suggested that brain-like computers can be realized by combining molecular circuits and machine leaning [48]. Furthermore, quantum interference has been proposed to be important for making devices [71]. Therefore, for the rational design of high-performance molecular circuits, we need to determine the relationship between the open-shell nature, electron conductivity, and quantum interference.

In this study, we found a possibility that the quantum interference effect in the molecular circuit can also be controlled by chemical substituents using the open-shell nature. To the best of our knowledge, this is the first report that discusses a relationship between quantum interference and the open-shell nature.

## Figures and Tables

**Figure 1 nanomaterials-14-00098-f001:**
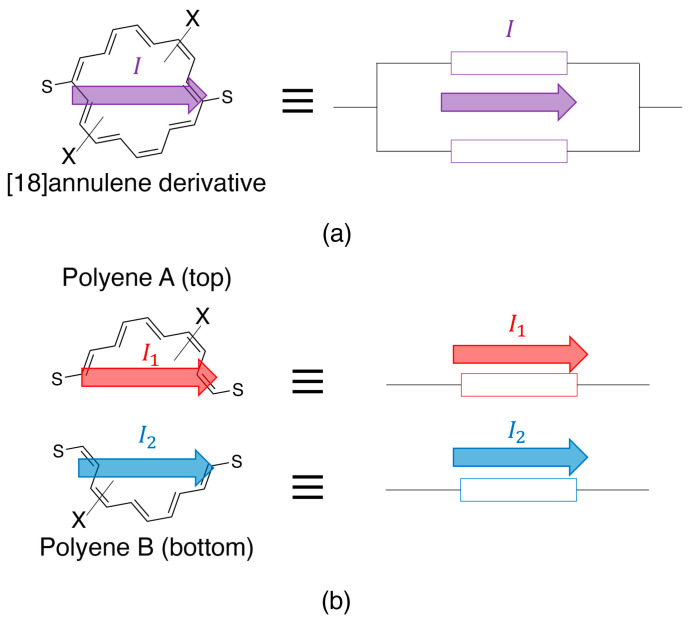
Molecular parallel circuit models: (**a**) [18]annulene derivative as the parallel circuit, and (**b**) two polyenes divided from the [18]annulene derivative.

**Figure 3 nanomaterials-14-00098-f003:**
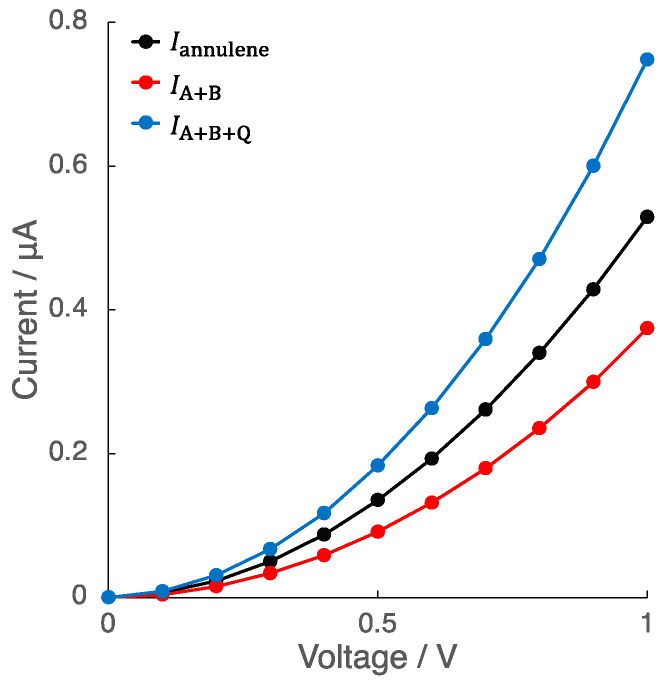
*I*–*V* characteristics of the [18]annulene ($I_{\mathrm{annulene}}$), the sum of independent polyenes ($I_{A+B}$), and the values including the quantum interference (QI) term ($I_{A+B+Q}$).

**Figure 4 nanomaterials-14-00098-f004:**
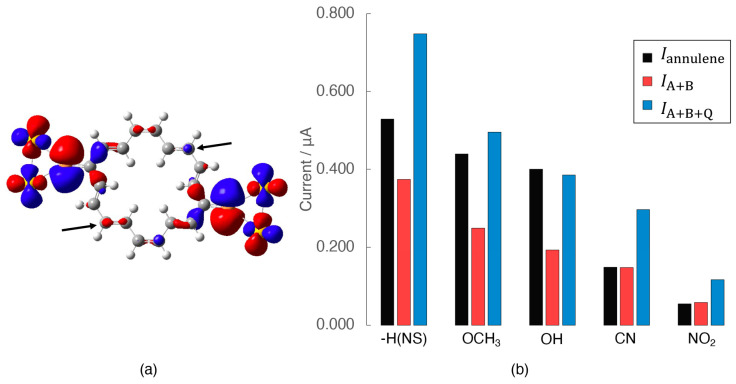
(**a**) Molecular orbital distribution of HOMO−5 of the [18]annulene extended molecule. The isovalue of the distribution is 0.02. Substituents are introduced at the carbons indicated by arrows. (**b**) Current of the [18]annulene derivatives ($I_{\mathrm{annulene}}$) and the divided polyenes ($I_{A+B}=I_{\mathrm{Polyene}\;A}+I_{\mathrm{Polyene}\;B}$, $I_{A+B+Q}=I_{\mathrm{Polyene}\;A}+I_{\mathrm{Polyene}\;B}+2\sqrt{I_{\mathrm{Polyene}\;A}I_{\mathrm{Polyene}\;B}}$) at 1.0 V.

**Figure 5 nanomaterials-14-00098-f005:**
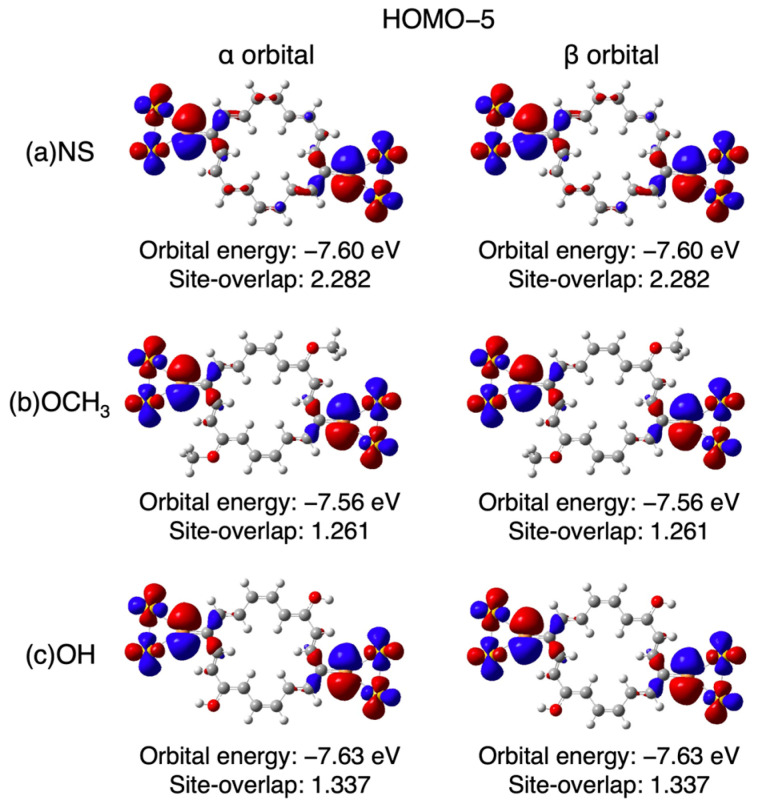
Electron distribution in HOMO−5 of (**a**) NS annulenes, (**b**) OCH_3_-substituted annulenes, and (**c**) OH-substituted annulenes, where the isovalues are 0.02. Other molecular orbitals of all models are summarized in Appendix A.

**Table 1 nanomaterials-14-00098-t001:** Summary of $\langle {\hat{S}}^{2}\rangle$ and coupling constants of the [18]annulene and the divided independent polyenes. In this table, α and β represent α and β orbitals, respectively. Other parameters related to the transition probability, such as orbital energy, Fermi energy, and so on, are shown in Appendix A.

	[18]Annulene		Polyene A		Polyene B	
	α	β	α	β	α	β
$\langle {\hat{S}}^{2}\rangle$	0.000		0.147		0.147	
$\gamma_{L1,\sigma }/eV$	0.855	0.855	1.078	0.674	0.675	1.049
$\gamma_{NR,\sigma }/eV$	0.855	0.855	0.675	1.049	1.078	0.674

**Table 2 nanomaterials-14-00098-t002:** Site-overlap values of each orbital for the [18]annulene and their divided independent polyenes. Total values of each orbital are also indicated in the same table.

	Site-Overlap					
	[18]Annulene		Polyene A		Polyene B	
	α	β	α	β	α	β
HOMO−9	0.0000	0.0000	0.0002	0.0000	0.0001	0.0000
HOMO−8	0.0213	0.0213	0.0000	0.0002	0.0000	0.0001
HOMO−7	0.0158	0.0158	0.0000	0.0000	0.0000	0.0000
HOMO−6	0.4074	0.4074	0.0000	0.0000	0.0000	0.0000
HOMO−5	2.2823	2.2823	0.2068	0.1903	0.1287	0.1534
HOMO−4	0.9860	0.9860	0.0043	1.5019	0.0036	1.4478
HOMO−3	0.0033	0.0033	0.0557	0.9201	0.0465	0.8951
HOMO−2	0.0150	0.0150	0.0015	0.0035	0.0000	0.0023
HOMO−1	0.0321	0.0321	0.0000	0.0000	0.0000	0.0000
HOMO	0.0331	0.0331	0.0000	0.0000	0.0000	0.0000
LUMO	0.0703	0.0703	0.0578	0.0351	0.0453	0.0515
LUMO+1	0.0341	0.0341	0.0002	0.0001	0.0000	0.0000
LUMO+2	0.0001	0.0001	0.0099	0.0013	0.0022	0.0001
LUMO+3	0.0023	0.0023	0.0308	0.0323	0.0342	0.0593
LUMO+4	0.0003	0.0003	0.0151	0.0112	0.0127	0.0158
LUMO+5	0.0037	0.0037	0.0000	0.0000	0.0030	0.0027
LUMO+6	0.0130	0.0130	0.0003	0.0002	0.0000	0.0000
LUMO+7	0.0000	0.0000	0.0135	0.0128	0.0310	0.0294
LUMO+8	0.0001	0.0001	0.0016	0.0016	0.0018	0.0017
LUMO+9	0.0170	0.0170	0.0000	0.0000	0.0000	0.0000
Total	3.9372	3.9372	0.3976	2.7105	0.3092	2.6593

**Table 3 nanomaterials-14-00098-t003:** $\langle {\hat{S}}^{2}\rangle$ values of each extended molecule.

Substituent X	$\langle {\hat{S}}^{2}\rangle$		
	[18]Annulene	Polyene A	Polyene B
-H (NS)	0.000	0.147	0.147
-OCH_3_	0.000	0.163	0.163
-OH	0.000	0.156	0.156
-CN	0.462	0.222	0.222
-NO_2_	0.438	0.193	0.193

**Table 4 nanomaterials-14-00098-t004:** Summary of the coupling constants and total of Site-overlap ^(i),(ii)^ of each of the extended molecules. (**a**) OCH_3_-substituted, (**b**) OH-substituted, (**c**) CN-substituted, and (**d**) NO_2_-substituted molecules. α and β in the table represent α and β orbitals, respectively.

(a)	-OCH_3_					
	Annulene		Polyene A		Polyene B	
	α	β	α	β	α	β
$\gamma_{L1,\sigma }$/eV	0.847	0.847	0.474	0.880	0.983	1.283
$\gamma_{NR,\sigma }$/eV	0.847	0.847	1.283	0.983	0.880	0.474
Site-overlap ^(i)^	3.05	3.05	0.293	0.475	0.324	0.280
(**b**)	**-OH**					
	**Annulene**		**Polyene A**		**Polyene B**	
	**α**	**β**	**α**	**β**	**α**	**β**
$\gamma_{L1,\sigma }$/eV	0.832	0.832	0.448	0.828	1.287	1.001
$\gamma_{NR,\sigma }$/eV	0.832	0.832	1.287	1.001	0.448	0.828
Site-overlap ^(i)^	3.07	3.07	0.174	0.386	0.204	0.288
(**c**)	**-CN**					
	**Annulene**		**Polyene A**		**Polyene B**	
	**α**	**β**	**α**	**β**	**α**	**β**
$\gamma_{L1,\sigma }$/eV	0.497	0.991	1.007	0.602	0.911	0.517
$\gamma_{NR,\sigma }$/eV	0.991	0.497	0.518	0.911	0.602	1.007
Site-overlap ^(i)^	2.31	2.59	1.245	0.892	0.999	1.141
(**d**)	**-NO_2_**					
	**Annulene**		**Polyene A**		**Polyene B**	
	**α**	**β**	**α**	**β**	**α**	**β**
$\gamma_{L1,\sigma }$/eV	0.890	0.481	1.070	0.632	0.710	0.441
$\gamma_{NR,\sigma }$/eV	0.481	0.890	0.441	0.711	0.632	1.070
Site-overlap ^(i)^	1.04	0.896	0.380	0.523	0.558	0.296

(i) Sum of HOMO−9 and LUMO+9. (ii) The small differences in Site-overlap between Polyenes A and B are considered to originate from the convergence accuracy of the molecular orbitals (here, 10^−7^ a.u. in electron density).

## Data Availability

The data presented in this study are available on request from the corresponding author.

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
