# Peer review of "Theoretical Study on the Open-Shell Electronic Structure and Electron Conductivity of [18]Annulene as a Molecular Parallel Circuit Model"

_nanomaterials, 2023, doi:10.3390/nano14010098_

Round 1
Reviewer 1 Report
Comments and Suggestions for Authors
1. Figure 5, the isovalue surfaces of HOMO orbitals for OCH3 and OH look the same in the current paper, please improve.
2. What are the main motivations to study [18] annulene and breaking the ring into the two polyenes derivatives, as shown in Figure 1 in the paper. Specifically, what are the typical applications of a circuit in Figure 1(a), and also the Figure 1(b). Is the quantum interference a hazardous contribution?
3. What is the E-field strength in the electric conductivity calculation for the molecular circuit? The Ohm's law is only expected at the low E-field value, it is not judged simply from the applied voltage. The profiles presented in Figure 3 are all deviated from the linear Ohm's law, is it because the applied E-field strength already very high?
4. Figure 4(b) shows some interesting results about suppressing the quantum interference term in the current, so what are the implications for the practical designing, or the use of molecular circuit models proposes in this paper.
Author Response
Response to Reviewer 1
Thank you very much for your valuable comments. We revised the manuscripts along your comments. We really appreciate your comments.
- Figure 5, the isovalue surfaces of HOMO orbitals for OCH3 and OH look the same in the current paper, please improve.
Thank you for your suggestion. We confirmed that the HOMOs for OCH3 and OH were surely different. In order to clarify the differences and to avoid the confusion, we identified them by “(a) NS, (b)OCH3, (c)OH “ in Figure 5.
- What are the main motivations to study [18] annulene and breaking the ring into the two polyenes derivatives, as shown in Figure 1 in the paper. Specifically, what are the typical applications of a circuit in Figure 1(a), and also the Figure 1(b). Is the quantum interference a hazardous contribution?
Thank you very much for your comment. Indeed, the motivation and application of this study are quite important. There are two motivations in this study. As already well known, molecular devices are one of the targets of nanomaterials. Candidates of molecular components have been proposed, however, there have been no guidelines to make “circuits” based on the molecular components. This is the main motivation of our study. As a realistic models of those molecular circuits, we consider that a larger π conjugated ring systems can be the good model for a parallel molecular circuit consists of two π conjugated chains. For the reason, we focused on [18]annulene as a stable but larger π conjugated ring. In addition, a relationship between the open-shell nature, electron conductivity and quantum interference has not been elucidated. Those are our motivation of this study.
Concerning the application, information processing is considered. For example, it is suggested that the brain-like computers can be realized by combining the molecular circuit and machine leaning (48. Tanaka et al).
On the other hand, the quantum interference was suggested to be important for making devices (Okuno, Y.; Ozaki, T. First-Principles Study of Multiterminal Quantum Interference Controlled Molecular Devices. J. Phys. Chem. C 2013, 117, 100-109. https://doi.org/10.1021/jp309455n). For the rational design of the molecular circuit, therefore, we have to know a relationship between the open-shell nature, electron conductivity and quantum interference.
From above points of view, we added sentences and a reference in the conclusion.
“Recently, it is suggested that the brain-like computers can be realized by combining the molecular circuit and machine leaning [48]. Furthermore, the quantum interference was suggested to be important for making devices [67]. For the rational design of the high-performance molecular circuit, therefore, we need to know a relationship between the open-shell nature, the electron conductivity and the quantum interference.”
- Okuno, Y.; Ozaki, T. First-Principles Study of Multiterminal Quantum Interference Controlled Molecular Devices. J. Phys. Chem. C 2013, 117, 100-109. https://doi.org/10.1021/jp309455n”
- What is the E-field strength in the electric conductivity calculation for the molecular circuit? The Ohm's law is only expected at the low E-field value, it is not judged simply from the applied voltage. The profiles presented in Figure 3 are all deviated from the linear Ohm's law, is it because the applied E-field strength already very high?
Thank you for your comments. Indeed, the point you suggested is not clearly explained in the text. In this paper, we estimate the I-V characteristics by considering transition probabilities (eq. 13) based on tunneling through the discrete molecular orbitals and their energies. In order to avoid misleading, we add sentences in section 2.1 as follows.
“In this way, the current can be calculated by considering transition probabilities based on tunneling through the discrete molecular orbitals and their energies without an external electric field.”
- Figure 4(b) shows some interesting results about suppressing the quantum interference term in the current, so what are the implications for the practical designing, or the use of molecular circuit models proposes in this paper.
Thank you for your comments. As answered above, the quantum interference is important issue for the consideration of the molecular devices. Our results strongly suggest that the quantum interference can be controlled by the introduction of the electron-donating/withdrawing groups into π conjugated systems. This information would be a design guideline for the molecular devices. To emphasize this point, we added below sentences in the conclusion.
“For the rational design of the high-performance molecular circuit, therefore, we need to know a relationship between the open-shell nature, the electron conductivity and the quantum interference.”
Reviewer 2 Report
Comments and Suggestions for Authors
In this work, the authors have examined the electron conductivities of [18]annulene and its derivatives as a molecular parallel circuit model consisting of two linear polyenes. By using elastic scattering Green’s function (ESGF) theory and density functional theory (DFT) methods, their electron conductivities are estimated. By the introduction of electron withdrawing groups into the annulene framework, a spin-polarized electronic structure appears and the quantum interference effect is significantly suppressed. Total current is also affected by the spin-polarization. This work is of interest to other researchers in scientific and engineering community of molecular electronics. However, there are a few comments to be addressed. The detailed comments are as follows:
1. In introduction, the authors write: “Such a miniaturization technology of transistors has increased integration density, which drastically improves the performance of electron devices. On the other hand, it is suggested that the miniaturization of the silicon-based devices is almost reaching to its limit. In addition, the energy loss due to a leakage current with the miniaturization has also been a problem.” The introduction has much room to be improved. Besides silicon-based devices, miniaturization technology of GaN-based devices has increased integration density, which drastically improves the performance of devices. To give the readers a much broader view, recent developments related to GaN-based devices, such as Nature Electronics 5, 723–734 (2022) https://doi.org/10.1038/s41928-022-00860-5; Optics Express 27(12), A669 (2019) https://doi.org/10.1364/OE.27.00A669; Optics Express 25(22), 26615 (2017) https://doi.org/10.1364/OE.25.026615, etc. should be added, so that the readers can be clear about the state-of-the-art of this topic.
2. The authors give the Table 1 in page 7 without any explanations. However, what can we learn from Table 1?
3. Figure 5 only shows electron distribution in HOMO-5 of donating groups. Why did not show the electron distribution in HOMO-5 of electron withdrawing groups?
4. The electron conductivity of the single-molecule is calculated. If it's multiple molecules, how to get the electron conductivity?
5. Some mistakes are found. For example, in line 224, “From the calculated ⟨𝑺2⟩ values, the [18]annulene is found to be a closed-shell system, whereas the divided polyenes with are the weak open-shell systems.”. Please proofread the manuscript carefully.
Comments on the Quality of English Language
This paper requires a grammar and spell check.
Author Response
Response to Reviewer 2
Thank you very much for your valuable comments. Especially, your comment for recent GaN-based devices improved our manuscript. We revised the manuscripts along your comments. We really appreciate your comments.
- In introduction, the authors write: “Such a miniaturization technology of transistors has increased integration density, which drastically improves the performance of electron devices. On the other hand, it is suggested that the miniaturization of the silicon-based devices is almost reaching to its limit. In addition, the energy loss due to a leakage current with the miniaturization has also been a problem.” The introduction has much room to be improved. Besides silicon-based devices, miniaturization technology of GaN-based devices has increased integration density, which drastically improves the performance of devices. To give the readers a much broader view, recent developments related to GaN-based devices, such as Nature Electronics 5, 723–734 (2022) https://doi.org/10.1038/s41928-022-00860-5; Optics Express 27(12), A669 (2019) https://doi.org/10.1364/OE.27.00A669; Optics Express 25(22), 26615 (2017) https://doi.org/10.1364/OE.25.026615, etc. should be added, so that the readers can be clear about the state-of-the-art of this topic.
Thank you very much for your comment. We apologize that our manuscript did not mention about a recent development of the GaN-type devices. Indeed, we must touch about those devices.
In the manuscript, therefore, we added below sentences including references you suggested.
“Recent developments of fabrication technology of power semiconductors using silicon carbide (SiC) and gallium nitride (GaN) have succeeded in realizing the higher efficiency and lower power consumption than those of the Si-based semiconductors [4-6].”
- The authors give the Table 1 in page 7 without any explanations. However, what can we learn from Table 1?
Thank you for your suggestion. We aimed to explain that there are differences in coupling constants and site-overlap between open- and closed-shell electronic states based on <S2> in Table 1. As you suggested, however, the explanation is not be enough. Therefore, we emphasized the point in the main text as follows.
“From the calculated values in Table 1, the [18]annulene is found to be a closed-shell system, whereas the divided polyenes are the weak open-shell systems. Moreover, according to Table 1 and Table 2, the coupling constants and Site-overlap values of a and b orbitals are same in the [18]annulene, while they are different in the divided polyenes.”
In addition, the HOMO-LUMO gap values were move to SI because they were not touched in the main text. With the same reason, the HOMO-LUMO gap values in Table 4 were also move to SI.
- Figure 5 only shows electron distribution in HOMO-5 of donating groups. Why did not show the electron distribution in HOMO-5 of electron withdrawing groups?
Thank you very much for your comment. Indeed, our explanation is not enough. We only showed HOMO-5 of models with electron donating groups in the main text because we focused on why the electron donating groups decrease the electron conductivity. In the case of the electron donating groups, the distribution of HOMO-5 is important rather than the spin polarization. In the case of the withdrawing groups, however, we can explain the decrease in the conductivity only by the spin polarization and molecular orbitals do not affect so much. That is a reason why we only discuss HOMO-5 of the electron donating groups in the main text. In order to avoid the misleading, we added sentences below.
“On the contrary, a difference in molecular orbital distribution, especially HOMO-5, becomes important for the models with the electron donating group. The distribution in HOMO-5 of NS model and the electron donating groups substituted annulenes are shown in Figure 5. By introducing the electron donating groups, a degree of delocalization of HOMO-5 that contributes to the conductivity is suppressed.”
And other orbitals (HOMO-2 – HOMO-5) of all models are depicted in Figure S2 in SI.
- The electron conductivity of the single-molecule is calculated. If it's multiple molecules, how to get the electron conductivity?
Thank you for your question. In the case of systems consist of the discrete orbitals, our approach is applicable. For the larger systems with continuous energy band structure, however, we must perform the band calculations. To emphases the point, we modified the sentence as follows.
“In this study, electron conductivity of the single-molecule, which has discrete orbitals, is calculated with the elastic scattering Green’s function proposed by Mujica and Luo [53].”
- Some mistakes are found. For example, in line 224, “From the calculated ⟨?2⟩values, the [18]annulene is found to be a closed-shell system, whereas the divided polyenes with are the weak open-shell systems.”. Please proofread the manuscript carefully.
We apologize the typo and some errors. We carefully corrected the errors.
Reviewer 3 Report
Comments and Suggestions for Authors
Despite the approximations used, the manuscript brings interesting results on the title topics. In future studies, authors might test their treatment on A and B polyenes with different lengths or of opposite charges. Gaussian enables calculations in electric fields as well.
My comments can be summarized as follows:
i) Ad Section 2.1: The authors should mention that the equations analogous to (14)-(16) are valid for the right electrode as well.
ii) The charges and spin states of the studied systems must be presented in Section 2.2.
iii) Line 206: ‘(𝐼Polyene A + 𝐼Polyene B : 𝐼A+B)’ should read ‘(𝐼Polyene A + 𝐼Polyene B = 𝐼A+B)’. Analogously at line 208.
iv) HOMO-LUMO gap units are missing in Table 1.
v) Although the A and B polyenes are equal, they differ in γ (Tables 1 and S2), site-overlap (Tables 2 and S2) and d (Table S2) values. The accuracy of the methods used should be discussed.
vi) What is the MO isosurface value in Figure 4?
vii) I propose to present HOMO-5 for A and B polyenes in Supplementary.
viii) Several misprints and grammatical errors can be found by a more careful reading.
Comments on the Quality of English Language
Minor corrections demanded.
Author Response
Response to Reviewer 3
Thank you very much for your valuable comments. We revised the manuscripts along your comments. We really appreciate your comments.
- i) Ad Section 2.1: The authors should mention that the equations analogous to (14)-(16) are valid for the right electrode as well.
Thank you very much for your comment. Indeed, the equations (15)-(17) (the referee perhaps suggests equations (15)-(17) but not (14)-(16)) are valid for the right electrode. So, we added sentences below.
“The equation (15) – (17) can be applicable to the interaction between the molecule and right electrode with subscript R and N, as well.”
- ii) The charges and spin states of the studied systems must be presented in Section 2.2.
Thank you for your suggestion. We present charge and spin states in Section 2.2 as
“For all models, charge and spin states were assumed to be neutral and singlet, respectively.”
iii) Line 206: ‘(?Polyene A + ?Polyene B : ?A+B)’ should read ‘(?Polyene A + ?Polyene B = ?A+B)’. Analogously at line 208.
Thank you very much. We revised it.
- iv) HOMO-LUMO gap units are missing in Table 1.
We apologize that we did not clarify the unit. We added the unit. (The HOMO-LUMO gap values were moved to SI as referee 2’s suggestion.)
- v) Although the A and B polyenes are equal, they differ in γ (Tables 1 and S2), site-overlap (Tables 2 and S2) and d (Table S2) values. The accuracy of the methods used should be discussed.
Thank you very much for your comment. Actually, as you indicated, there are small asymmetry between a and b orbitals in γ and d values for NS, OH and CN in a range of 0.01 – 0.04. We increased the convergence criteria of the DFT calculations then we obtained the symmetric values as summarized in the revised manuscript. However, asymmetry still exists in Site-overlap. This originates in the negligible asymmetry involved in the molecular orbitals within the convergence (10–7 a.u. in electron density). Therefore, we added sentences below in caption of Figure 4. (Figure numbers are different from the previous manuscript).
“b) The small differences in Site-overlap between Polyenes A and B are considered to originate from the convergence accuracy of the molecular orbitals ( Here 10–7 a.u. in electron density).”
- vi) What is the MO isosurface value in Figure 4?
We apologize that we did not mention about the isosurface value. The value is 0.02. So, we added the isosurface value in the Figure5 cations.
vii) I propose to present HOMO-5 for A and B polyenes in Supplementary.
Thank you for your suggestion. We presented HOMO-2 to HOMO-5 of all model structures including polyenes A and B in SI.
viii) Several misprints and grammatical errors can be found by a more careful reading.
We apologize the typo and some errors. We carefully corrected the errors.